# The Role of HER2 Status in the Biliary Tract Cancers

**DOI:** 10.3390/cancers15092628

**Published:** 2023-05-05

**Authors:** Ruveyda Ayasun, Muhammet Ozer, Ilyas Sahin

**Affiliations:** 1Laura and Isaac Perlmutter Cancer Center, New York University Langone Medical Center, New York, NY 10016, USA; ruveyda.ayasun@nyulangone.org; 2Department of Medical Oncology, Dana Farber Cancer Institute, Harvard Medical School, Boston, MA 02132, USA; muhammet_ozer@dfci.harvard.edu; 3Division of Hematology/Oncology, Department of Medicine, University of Florida Health Cancer Center, Gainesville, FL 32608, USA

**Keywords:** biliary tract cancer, HER2 status, precision medicine

## Abstract

**Simple Summary:**

Human epidermal growth factor receptor-2 (HER2) plays a critical role in breast and gastric cancer. More recently, a growing body of literature demonstrated that HER2 is involved in biliary tract cancers. In this review, we discuss the potential therapeutic utilization of HER2-blocking antibodies and ongoing clinical trials assessing the efficacy of HER2-blocking antibodies in HER2-altered and/or -amplified biliary tract cancers.

**Abstract:**

Despite recent advances, biliary tract cancer (BTC) is traditionally known as being hard to treat with a poor prognosis. Recent state-of-the-art genomic technologies such as next-generation sequencing (NGS) revolutionized cancer management and shed light on the genomic landscape of BTCs. There are ongoing clinical trials to assess the efficacy of HER2-blocking antibodies or drug conjugates in BTCs with HER2 amplifications. However, HER2 amplifications may not be the sole eligibility factor for these clinical trials. In this review, we aimed to comprehensively examine the role of somatic HER2 alterations and amplifications in patient stratification and provide an overview of the current state of ongoing clinical trials.

## 1. Introduction

Biliary tract cancers (BTCs) represent 3% of all gastrointestinal cancers and have a dismal prognosis with a 5 year survival rate of 2% [1]. BTCs consist of intrahepatic cholangiocarcinoma, extrahepatic cholangiocarcinoma, gallbladder cancer, and ampulla of Vater cancer [2]. Geographical location is one of the determinants of the incidence of BTCs. Due to endemic liver flukes, cholangiocarcinoma is more common in southeast Asian countries [3]. Although a growing body of literature demonstrated the efficacy of targeted therapies in patients with BTC, systemic chemotherapy remained the backbone treatment for this devastating disease. Gemcitabine with cisplatin (GEMCIS) is still recommended as first-line treatment (ABC-02 trial) [4] and mFOLFOX with active symptom control as second-line therapy (ABC-06 trial) [5]. In an era of immune-based therapies, the TOPAZ-1 trial evaluated the efficacy of durvalumab (anti-PDL1) plus GEMCIS in patients with BTC (NCT03875235). The addition of durvalumab to standard of care treatment significantly improved overall survival rate in patients with BTC. The estimated 24 month overall survival rate (OS) was 24.9% and 10.4% in the durvalumab plus GEMCIS and placebo plus GEMCIS arm, respectively [5]. Median OS was 12.8 months in the durvalumab plus GEMICS group and 11.5 months in the GEMCIS group. Considering the observed survival advantage associated with the addition of durvalumab, the NCCN updated their guidelines and reclassified GEMCIS plus durvalumab as the preferred category 1 recommendation for BTC. Despite the increased survival advantage, the disease prognosis remained dismal and the objective response rate with the addition of durvalumab to GEMCIS improved objective response rates from 18.7% to 26.7%. As a second-line treatment, options are limited, and the prognosis is very poor for patients without access to targeted therapy. The median OS for mFOLFOX as a second-line treatment showed only a minimal survival benefit of 6.2 months compared to 5.3 months in active symptom control, with an objective response rate (ORR) of 5%. These findings highlighted an unmet need for the development of novel treatments, not only as first-line therapy but also for the management of treatment-refractory BTC [5].

BTCs are heterogenous in terms of anatomy, thus their pathogenesis and genomic drivers may vary. Despite precision medicine having revolutionized cancer care over the last decades, there is an unmet need for more precise approaches in patients with BTC. The number of somatic alterations significantly differs in patients with BTC based on primary site of tumor. For instance, there is a significant difference between intrahepatic cholangiocarcinoma and gallbladder cancer or extrahepatic CCA in terms of somatic alterations [1]. To date, fibroblast growth factor receptor (FGFR) and isocitrate dehydrogenase (IDH) have been extensively studied in intrahepatic biliary tract cancers [2]. According to the results of a phase 2 study, the median overall survival (OS) data for the cohort of patients with FGFR2 fusion/rearrangement who received pemigatinib as a second-line treatment had not yet been reached at the time of the data cutoff, which was 21.1 months. This suggests that the survival benefit of pemigatinib for this patient population may be substantial, although longer follow-up is needed to confirm these results [6]. However, for the cohorts with other FGFR alterations and no FGFR alterations, the median OS was 6.7 months and 4.0 months, respectively. The 12 month OS rate for the cohort with FGFR2 fusion/rearrangement was 68%. Consequently, pemigatinib, an FGFR1-3 inhibitor, received FDA approval for the treatment of patients with BTC with FGFR2 fusions in April 2020 [6]. After the approval of a FGFR1-3 inhibitor by the FDA for the treatment of BTC, several other FGFR inhibitors were tested for their effectiveness. As a result, clinical trials were conducted to evaluate the efficacy of infigratinib and futibatinib. In a phase 2 trial evaluating infigratinib, another competitive and reversible inhibitor of FGFR1-3, as a second-line treatment option, objective response rate was determined to be 23.1%, while the median progression free survival (PFS) was 7.3 months [7]. Thus, infigratinib was granted accelerated approval by FDA in 2021 for previously treated, unresectable locally advanced, or metastatic cholangiocarcinoma with FGFR2 rearrangements. Futibatinib, a selective and irreversible FGFR1–4 inhibitor, obtained accelerated FDA approval in September 2022 after the completion of the FOENIX-101 phase 1 trial that enrolled 86 patients [8]. The results demonstrated partial responses in five patients, including three with FGFR2 fusion-positive intrahepatic cholangiocarcinoma and two with FGFR1-mutant primary brain tumors. Additionally, 41 patients (48%) achieved stable disease. Following the completion of the FOENIX-101 study, the efficacy of futibatinib was evaluated in the FOENIX-CCA2 phase 2 trial. A total of 42% of patients (95% CI, 32 to 52) had a response, and the median duration of response was 9.7 months. After a median follow-up period of 17.1 months, the trial showed a median PFS of 9.0 months and an OS of 21.7 months [9].

Isocitrate dehydrogenase 1 (IDH1) alterations have been found in approximately 20% of patients with intrahepatic cholangiocarcinoma. In a phase 3 trial (ClarIDHy), it has been demonstrated that ivosidenib yielded a favorable median OS benefit with a tolerable safety profile when compared to the placebo group (10.3 months 95% CI, 7.8–12.4 months and 5.1 months 95% CI, 3.8–7.6 months, respectively) [10].

The prevalence of BRAFV600E mutations, another actionable target for BTC, is approximately 3% [11]. BRAFV600E mutation in biliary tract cancer has been linked to worse long-term OS than patients without this mutation. In phase 2 clinical trial (ROAR), dabrafenib plus trametinib has been evaluated in patients with BTC with BRAFV600E mutation with encouraging results [12]. The overall response (OR) was achieved by 47% of patients (95% CI 31–62).

Given aforementioned milestones in this hard-to-treat cancer, the utilizing of genomic profiling for all patients is crucial to guide treatment decisions and patient enrolment into clinical trials. In the MOSCATO 01 trial, the benefit of utilizing next-generation sequencing (NGS) in the management of patients with BTC has been demonstrated. Remarkably, tumor biopsy and NGS was utilized in 843 patients, and actionable targets were identified in 411 out of 843 patients [13]. Actionability has been described in the ESMO guidelines to raise awareness among clinicians. IDH1 and FGFR2 mutations in this guideline have been classified as ESCAT I (ready for routine implementation). Moreover, HER2 alterations have been classified as ESCAT III, implying that mutation–drug match is suspected to improve outcomes in other cancer types [1,14]. Galdy et al. published a meta-analysis in 2016 which showed that the prevalence of HER2 overexpression varied depending on the anatomic origin of the biliary tract cancer. The study found that extrahepatic cholangiocarcinoma had a higher rate of HER2 overexpression (17.4%) compared to intrahepatic cholangiocarcinoma (4.8%) [15].

In this review, we aim to critically evaluate the rationale and status of targeting HER2 and what the future holds for this class of cancer therapeutics in patients with advanced or unresectable HER2-positive BTC.

## 2. Prognostic Role of HER2 Status in BTC

There are controversial studies pertaining to the prognostic role of HER2 alterations in patients with BTC. A retrospective study of the 100 resected BTC cases from Italy found that 11% of cases exhibited HER2 overexpression, which was defined as a score of 3+ by IHC or a score of 2+ that was confirmed by FISH amplification. In the study, disease-free survival (DFS) was significantly shorter in patients with HER2-positive BTC compared to HER2-negative patients with 10.6 and 20.9 months, respectively. Although there was a noticeable disparity in the median overall survival between HER2-positive and -negative patients, which was 23.4 months and 55.2 months, respectively, the difference did not achieve statistical significance (*p* = 0.068) [16]. The median overall survival (OS) in various types of biliary tract cancers (BTCs) differed significantly, as indicated by this study. In particular, ampullary tumors had not yet reached the median OS, intrahepatic tumors had a median OS of 55.3 months, extrahepatic tumors had a median OS of 34.7 months, and gallbladder tumors had a median OS of 18.1 months. It is worth noting that these differences suggest that the primary site of BTC may be an important factor in determining patient prognosis. In another study from South Korea, HER2 alterations have been detected in 14.9% of patients with advanced BTC [17]. The patients with and without HER2 aberrations did not have a significant difference in tumor response to GEMCIS in the study (33.3% vs. 26.2%, *p* = 0.571). The median progression-free survival (PFS) to GP was 4.7 months (95% CI, 4.0 to 5.5 months) for patients with HER2 aberrations and 7.0 months (95% CI, 5.2 to 8.8 months) for those without HER2 aberrations (*p* = 0.776). Moreover, the median OS was not reached in either group (*p* = 0.739). The role of HER2 aberrations as an independent biomarker was evaluated through univariate analyses for PFS to GEMCIS and OS. The analysis for PFS to GEMCIS revealed that the grade of differentiation (poorly differentiated vs. well/moderately differentiated), disease stage (metastasis vs. locally advanced), and number of metastatic sites (≤2 vs. >2) were significant independent factors. However, HER2 aberrations did not demonstrate statistical significance as an independent factor [17]. The results of these studies indicated that there might be differences in the prevalence and characteristics of biliary tract cancers between various anatomical regions and between countries in Asia and non-Asia. Moreover, there are retrospective studies that investigated the differences in HER2 overexpression in Asian and Caucasian patients with biliary tract cancer. A meta-analysis revealed that HER2 expression was more prevalent in Asian patients (28.4%) than western patients (19.7%) [18]. In a study from Japan, 454 cases of biliary tract cancer have been assessed for HER2 positivity. It has been demonstrated that HER2 positivity differed among different subtypes of BTCs (3.7% in iCCA, 3% in perihilar eCCA, 18.5% in distal eCCA, 31.3% in GBC, and 16.4% in ampullary cancer) [19]. The percentage of HER2 overexpression observed in different studies may differ because of varying factors such as the use of different cutoff values to determine overexpression, the diverse detection methods employed, and the specific site of the primary tumor. Nevertheless, additional prospective cohorts are needed to validate these findings and to conclude whether HER2 alterations are prognostic for patients with advanced BTC. Given that there are accumulating clinical trials that evaluate HER2-targeting agents in patients with BTC, next-generation sequencing, along with IHC or FISH, may be required to stratify patients for anti-HER2 therapies. Intriguingly, not only membranous staining but also cytoplasmic staining might have an implication in the survival of patients with BTC. Ata et al. demonstrated that, though statistical significance was limited, lower cytoplasmic HER2 scores have been correlated with longer survival in patients with pancreatic, gallbladder, cholangiocarcinoma, and periampullary cancers (*p* = 0.052) [20].

## 3. HER2 Alterations in BTC

HER2-targeting agents have demonstrated remarkable responses in patients with HER2 alterations in gastric and breast cancer. Based on the ToGA study, HER2 scoring allows the appropriate selection of patients eligible for treatment with HER2-targeted therapies in gastric cancer. This study also revealed that immunohistochemistry should be the initial test and 2+ samples should be retested with FISH [21]. Hence, the vast majority of our knowledge on the role of HER2 in oncology stems from breast and gastric cancer trials. Due to the widespread acceptance of tumor-agnostic approaches in precision medicine, there is currently significant interest in investigating whether anti-HER2 therapies can be applied to other types of cancer, including biliary tract cancer (BTC). This may have important implications for the development of targeted therapies for BTC and other cancers with HER2 aberrations. It has been previously demonstrated that 54.3% of patients with BTC have a HER2 IHC score of 1+ and that 10.9% of them have a HER2 score of 3+ [2,22]. More recently, a nationwide retrospective study for clinicopathological data of 642 gallbladder cancer (GBC) patients from the Netherlands unveiled that about 50% of patients with GBC harbor actionable targets [23]. In this study, HER2 overexpression (IHC score of 3+) has been observed in 7% of patients with GBC. Notably, HER2 mutations typically occur in the absence of HER2 amplifications in all types of cancers [2]. G660D, V659E, R678Q, and Q709L mutations have been recognized as the most common HER2 alterations in all types of cancers. Functional analysis of these alterations has shown them to be activating [24]. Although HER2 amplifications (5–15%) and overexpression (20%) have been observed more frequently than HER2 mutations (2%) in BTC, activating HER2 mutations emerged as druggable by anti-HER2 therapies in patients with BTC without amplification or overexpression [2]. The frequency of HER2 alterations may vary according to the primary tumor site. Researchers from South Korea analyzed the HER2 status of 121 patients with BTC. HER2 alterations were found in 14.9% of patients with the highest frequency in GBC (36.4%) (Table 1) [17]. In addition to gene amplification or overexpression, point mutations are detected in this patient cohort as summarized in Table 2. These findings encourage future world-wide studies investigating HER2 aberrations in patients with BTC (Table 3). Since point mutations have been found in 27.8% of patients, it is critical to assess whether they are activating mutation, namely, druggable by HER2-targeting agents such as neratinib [25].

## 4. Resistance to Anti-HER2 Therapies

There are multiple FDA-approved HER2-targeted therapies, including antibody–drug conjugates (e.g., T-DM1 and DS-8201), monoclonal antibodies (e.g., trastuzumab and pertuzumab), and small-molecule HER1/2 TKIs (e.g., tucatinib, lapatinib, and neratinib) [27]. However, intrinsic or acquired resistance is a major limitation of these therapies. Activating HER2 mutations have been linked to resistance to lapatinib but responsiveness to neratinib in prior breast cancer studies [28]. The efficacy of neratinib, which is an irreversible HER2 inhibitor, has been evaluated in patients with HER2-mutant, nonamplified breast cancer. Ma et al. demonstrated in this trial that the clinical benefit ratio was 31% [29]. Therefore, an IHC score of 0 or 1+ HER2 may not be the sole predictor for responsiveness to HER2 therapies [30]. Due to intratumor heterogeneity, some cell clones within tumor tissue may express low levels of HER2 or may not be HER2-dependent, resulting in primary resistance to anti-HER2 therapies [31].

The downstream pathway of HER2 induces PI3K/AKT/mTOR activation, thus anti-HER2 therapies inhibit this signaling pathway. However, the constitutive activation of PI3K has previously been described in breast cancer patients. Alterations in PI3K/AKT/mTOR signaling pathway occur in 40% and 25% of extrahepatic cholangiocarcinoma (eCCA) and intrahepatic cholangiocarcinoma (iCCA) patients, respectively [32]. Hence, the clinical utility of PI3K inhibitors such as taselisib may benefit patients who are irresponsive to anti-HER2 therapies and have constitutively active PI3K signaling. Moreover, FGFR1 and FGF3 amplification has been linked to lower pathologic response in patients with HER2-positive breast cancer treated with neoadjuvant trastuzumab [33]. Given that pemigatinib (FGFR 1-2-3 inhibitor) was among the very first targeted therapies that was approved by the FDA for the management of advanced BTC [34], it could be considered in patients that progressed on anti-HER2 therapies. In the phase 2 SUMMIT basket trial, 25 patients with treatment-refractory metastatic BTC were enrolled and screened for HER2 mutations. The most common HER2 mutations in these cohorts were S310F (n = 11, 48%) and V777L (n = 4, 17%). In this trial, patients who received neratinib treatment demonstrated an ORR of 16% (95% CI, 4.5–36.1%) and a clinical benefit rate (CBR) of 28% (95% CI, 12.1–49.4%) [35]. The diagnostic tests for HER2 gene mutations that were approved by the FDA in August 2022 are as follows: Guardant360 CDx (blood) and Oncomine Dx Target Test (tumor tisssue). Even though HER2 amplifications (2–3%) or overexpression (2.5%) occur to a lesser extent in lung cancer, trastuzumab deruxtecan, an antibody–drug conjugate, received accelerated approval by FDA for patients with HER2-mutant non-small-cell lung cancer in August 2022 due to an encouraging phase 2 study [36]. In the study, trastuzumab deruxtecan showed durable responses in patients who had metastatic HER2-mutant non-small-cell lung cancer that was refractory to standard treatment. The confirmed objective response rate was 55% (95% CI, 44 to 65), with a median OS of 17.8 months (95% CI, 13.8 to 22.1). Objective responses were observed in various HER2 mutation subtypes and even among patients with undetectable HER2 expression or HER2 amplification. This landmark study may also provide the clinical rationale of HER2-targeted therapy for patients with HER2-mutant BTC.

## 5. Completed and Ongoing Clinical Trials

Several clinical trials have investigated the targeting of HER2 in BTC, with some reports available on their outcomes. Zanidatamab (ZW25) is a humanized bispecific antibody that targets both the juxtamembrane extracellular domain (ECD4) and the dimerization domain (ECD2) of HER2. This is similar to trastuzumab (T) and pertuzumab (P), which target the same domains, respectively. The anti-tumor activity of zanidatamab appears to be greater than trastuzumab both in vitro and in vivo, and across various HER2-overexpressing tumor cell types with different HER2 expression levels [37]. Findings from human cell line and animal models indicate that zanidatamab may also have clinical efficacy against tumors with low HER2 expression. The first in-human, multicenter, phase 1 trial reported the safety and tolerability of zanidatamab in patients with locally advanced or metastatic solid tumors, including BTC (intrahepatic cholangiocarcinoma (n = 5), extrahepatic cholangiocarcinoma (n = 4), and gallbladder cancer (n = 13)) with HER2 overexpression or amplification, who had received all available approved therapies [38] (NCT02892123). The study had 22 patients with HER2-overexpressing or -amplified BTC enrolled with a confirmed ORR of 38% (95% CI, 18–62) with a median duration of response of 8.5 months. These early-phase study results support that HER2 is a promising actionable target for treating BTC in patients with HER2 overexpression or amplification.

In the MyPathway study, 39 previously treated metastatic biliary tract cancers with HER2 amplification, HER2 overexpression, or both were treated with trastuzumab plus pertuzumab [39]. The study subgroups by tumor location within the biliary tract were intrahepatic (n = 7), extrahepatic (n = 7), gallbladder (n = 16), ampulla (n = 5), and undesignated (n = 4). In this patient population with a median follow-up of 8.1 months, 9 out of 39 patients (23%; 95% CI 11–39) achieved an ORR, while 20 patients (51%; 95% CI 35–68) achieved disease control rate. The median PFS was 4.0 months, median OS was 10.9 months, and the 1 year OS rate was 50% (NCT02091141). In the study, exploratory post hoc analyses were conducted on subgroups of patients with HER2 alterations. Among the six patients who had both HER2 amplification or overexpression and HER2 mutations, three showed partial response (S310F [n = 2] and T733I), one had stable disease (S310F), and two had progressive disease (T862A and H878Y). In four patients with HER2 amplification but without protocol-defined HER2 overexpression (IHC 3+), one showed partial response (IHC 2+), two had stable disease (both IHC 2+), and one had progressive disease (IHC 0). It is noteworthy that all of these four patients progressed within two months from treatment initiation.

A recent Korean single-arm phase 2 study evaluated trastuzumab plus FOLFOX regimen in patients with BTC harboring HER2 amplification or overexpression who are refractory to GEMCIS [40]. Out of the 34 patients in the study, 10 experienced a partial response, and 17 had stable disease, leading to an overall response rate of 29.4% (95% CI 16.7–46.3) and a disease control rate of 79.4% (95% CI 62.9–89.9). The median progression-free survival was 5.1 months (95% CI 3.6–6.7), and the median overall survival was 10.7 months (95% CI 7.9—not reached). Given the historical data demonstrating a 5% ORR with FOLFOX as second-line treatment for BTC [5], the promising results of this phase 2 study pave the way for future combination treatments utilizing HER2-targeted therapies.

There are ongoing clinical trials investigating HER2-targeting agents in BTC, such as zanidatamab alone (NCT04466891) and antibody–drug conjugates (ADCs) which are monoclonal antibodies chemically linked to a chemotherapeutic. The several ongoing clinical trials to assess the clinical activity of ADCs in a variety of cancers, including BTC, are listed in Table 4. The benefit of ADCs is to reduce off-target toxicity by binding specific tumor markers. In a phase 1 study, RC48-ADC showed promising activity in patients with HER2+ gastric cancer who were pretreated with HER-targeted drugs (ORR of 15% and DCR of 45%) [41]. Given that RC48-ADC demonstrated a tolerable safety profile and promising clinical activity in HER2+ gastric cancer, its efficacy in HER2+ unresectable BTCs is under investigation (NCT04329429). MRG002 is a type of investigational ADC that binds to HER2 on the surface of cancer cells. Subsequently, HER2-expressing cells internalize MRG002 and monomethyl auristatin E, which is conjugated to the HER2 antibody, blocks microtubule polymerization [42]. A phase 2 clinical trial is currently being conducted in China to evaluate the effectiveness of MRG002 for patients who have HER2-positive unresectable locally advanced or metastatic biliary tract cancer and who have experienced progression or relapse after their first-line treatment (NCT04837508). In a study from Japan, trastuzumab deruxtecan (T-Dxd), which is composed of a humanized monoclonal anti-HER2 antibody, a cleavable tetrapeptide-based linker, and a potent topoisomerase I inhibitor, has been tested in patients with HER2-positive BTC [43]. In the study, there were 22 patients with HER2-positive and 8 with HER2-low BTC. Among these 22 patients, 45.5% had IHC3+ and 54.5% had IHC2+/ISH+; the primary sites were gallbladder, extrahepatic, intrahepatic, and Vater in 11, 6, 3, and 2 patients, respectively. The confirmed ORR in HER2-positive patients was 36.4% (8/22; 2 complete response and 6 partial response; 90% CI, 19.6–56.1), which indicated a statistically significant improvement in ORR (*p* = 0.01). The disease control rate, median PFS, and median OS were 81.8% (95% CI, 59.7–94.8), 4.4 months (95% CI, 2.8–8.3), and 7.1 months (95% CI, 4.7–14.6), respectively. Encouraging efficacy was also observed in patients with HER2-low BTC, with an ORR of 12.5% (1/8; 1 partial response; 95% CI, 0.3–52.7), a disease control rate of 75.0% (95% CI, 34.9–96.8), a median PFS of 4.2 months (95% CI, 1.3–6.2), and a median OS of 8.9 months (95% CI, 3.0–12.8).

Given that trastuzumab deruxtecan received FDA approval for lung cancer, it seems feasible to test this ADC for BTC in the USA. Although HER2-mutant BTCs may not be as responsive to HER2 monoclonal antibodies as HER2-amplified or -overexpressed cancer types, irreversible HER2 inhibitors such as neratinib seem to be promising treatment options. HER2 ADCs are underway with the hope of overcoming resistance owing to the low HER2 expression. In a phase 3 trial for patients with HER2-low metastatic breast cancer who had received one or two previous lines of chemotherapy, T-Dxd has been shown to yield a survival benefit in patients with HER2-low advanced breast cancer compared to the physician’s choice of chemotherapy [44]. T-DXd has an advantage over other HER2-targeting due to its ability to induce a bystander effect. This is because T-DXd is internalized into cancer cells upon binding to HER2 and then releases its cytotoxic payload inside the cells, which can diffuse to neighboring cells and kill them even if they express lower levels of HER2. This makes T-DXd a promising treatment option for patients with heterogeneous HER2 expression within the tumor.

Lapatinib is a small molecule inhibitor that targets the ATP-binding pocket of the protein kinase domain in EGFR/HER2 receptors. In doing so, it prevents self-phosphorylation and blocks downstream signaling pathways, resulting in the inhibition of receptor-mediated signaling processes. Both EGFR and HER2/neu were thought to be potential targets for anticancer therapy in BTC as they were reported to be overexpressed in this cancer type and play a direct role in cholangiocarcinogenesis [45]. Based on the knowledge that both EGFR and HER2/neu may play a role in BTC, the hypothesis was that a dual inhibitor capable of blocking both targets would have a greater therapeutic benefit compared to compounds that only target one receptor. Consequently, two phase 2 studies were conducted to investigate the efficacy and tolerability of lapatinib, a single-agent inhibitor, in patients with advanced BTC [45,46]. Despite being widely used in the treatment of breast cancer, clinical trials using a single agent of lapatinib for biliary tract cancer have not produced positive results. These disappointing outcomes have led to the conclusion that targeting HER2 may not be an effective approach for managing biliary tract cancer. In a preclinical study, lapatinib demonstrated a synergistic effect in gemcitabine-responsive human gallbladder cells (TGBC1-TKB) when combined with gemcitabine [47]. In parallel to this study, lapatinib inhibited growth of HER2-overexpressing cholangiocarcinoma in patient-derived organoids and cell lines. Intriguingly, lapatinib in combination with gemcitabine demonstrated a synergistic effect via inhibiting the ABCB1 transporter which mediates resistance to gemcitabine in CCA patients [48]. Overall, it might be more feasible to stratify patients based on their initial HER2 expression and combine HER2-targeting drugs with other FDA-approved chemotherapeutics.

## 6. Conclusions—Concluding Remarks and Future Perspectives

Patients with BTC have a poor prognosis and, due to high heterogeneity, they are difficult to treat using common therapeutic strategies. However, this unique feature also presents an opportunity for personalized, targeted therapies. Novel targeted therapies such as IDH and FGFR inhibitors have revolutionized the management of BTC and, in recent years, various molecular aberrations have been identified, including HER2. Next-generation sequencing may help clinicians better stratify and predict which patients benefit from small molecule inhibitors the most. Moreover, the frequency of HER2 alterations may vary based on the anatomic localization of BTC. For instance, such alterations are more prevalent in gallbladder cancers than in other BTCs. To date, it is unclear whether the response to HER2-directed therapies may differ according to the primary tumor site. Clinical trials assessing the efficacy of HER2-targeted therapies in BTC pave the way for designing more precise clinical trials. As aforementioned, HER2 alterations are not limited to overexpression or amplification. Therefore, testing for HER2 alterations should not be limited to protein expression levels. BTC patients with activating point mutations of HER2 but without overexpression might benefit from novel HER2-targeted therapies such as neratinib or antibody-drug conjugates. Currently, durvalumab, combined with gemcitabine and cisplatin, is the new standard of care treatment for unresectable gallbladder cancer. As more clinical trials assess the efficacy of HER2-targeted therapies in BTC, it may change clinical practice in the near future. Collateral and vertical oncogenic signaling pathways are the current hurdle in precision medicine. Even though results from clinical trials assessing HER2-targeting therapies in BTC are promising, combinational treatment options might be imperative in achieving breakthrough responses. In this regard, the combining of immunotherapy or chemotherapy agents with anti-HER2 therapies seems to be feasible for greater and more durable responses [49]. The efficient development of targeted therapies and clinical trials for this rare and genomically heterogeneous disease will require collaboration between academic centers and industry to harmonize efforts.

## Figures and Tables

**Table 1 cancers-15-02628-t001:** The frequency of HER2 aberrations in South Korean patients with BTC (adapted from [17]).

Primary Tumor Site	Frequency of HER2 Alteration
Intrahepatic cholangiocarcinoma	5.8%
Extrahepatic cholangiocarcinoma	13.9%
Gallbladder cancer	36.4%
Ampulla of Vater	18.2%

**Table 2 cancers-15-02628-t002:** Subtypes of HER2 aberrations in South Korean patients with BTC (adapted from [17]).

Type of HER2 Alteration	Frequency of HER2 Alteration 100% (n = 18)
Point mutation	27.8% (n = 5)
Gene amplification	61.1% (n = 11)
Point mutation and gene amplification	11.1% (n = 2)

**Table 3 cancers-15-02628-t003:** HER2 aberrations in European patients with BTC [26].

Primary Tumor Site	Frequency of HER2 Alteration	Type of HER2 Alteration
Intrahepatic CCA	4.2%	Point mutation (23.8%) and copy number alteration (66.6%)
Extrahepatic CCA	9.7%	Point mutation (53.6%) and copy number alteration (41.2%)

**Table 4 cancers-15-02628-t004:** Clinical trials of HER2-targeted therapies in biliary tract cancers.

Clinical Trial	Drug	Patient Characteristics	Study Phase	Recruitment Status	Primary Endpoint	Results
NCT00478140	Trastuzumab	Previously treated, locally advanced or metastatic gallbladder cancer or bile duct cancer that cannot be removed by surgery	II	Terminated due to slow accrual	ORR, DOR	Terminated
NCT00107536	Lapatinib ditosylate	Previously treated, unresectable liver or biliary tract cancer (BTC)	II	Completed	ORR, PFS, AEs, median OS, OS	Completed
NCT04466891 (HERIZON-BTC-01)	Zanidatamab	Previously treated, advanced or metastatic HER2-amplified BTCs	II	Active, not recruiting	ORR, DOR, DCR, PFS, OS, AEs	Not recruiting
NCT04329429	RC48-ADC	Previously treated, locally advanced or metastatic HER2 overexpressed BTC who have failed first-line chemotherapy	II	Recruiting	ORR, DOR, PFS, OS, DCR, AEs	Active
NCT03929666	ZW25 (Zanidatamab) Capecitabine Cisplatin Fluorouracil Leucovorin Oxaliplatin Bevacizumab Gemcitabine	Previously treated, unresectable, locally advanced, recurrent, or metastatic HER2-expressing BTC	II	Recruiting	DLT, AEs, ORR	Active
NCT04837508	MRG002 ADC	Previously treated, unresectable locally advanced or metastatic BTC patients who have progressed during or relapsed after at least one prior stand therapy	II	Recruiting	ORR	Active
NCT04722133	Herzuma (Trastuzumab-pkrb) mFOLFOX	Previously treated, HER2-positive advanced/metastatic/nonresectable BTC	II	Recruiting	ORR, PFS, DCR, OS, AEs	Active
NCT05417230	RC48-ADC Envafolimab (anti-PDL1)	Previously treated, locally advanced or metastatic BTC with positive HER-2	II	Not yet recruiting	ORR, PFS, DCR, OS, AEs	Active
NCT04482309 (DESTINY-PanTumor02)	Trastuzumab Deruxtecan (T-DXd, DS-8201a)	HER2-overexpressing tumor-specific cohorts including BTC	II	Active, not recruiting	ORR, PFS, DCR, OS, AEs	Active, not recruiting
NCT05540483 (RIGHT)	RC-48 GLS-010 (anti-PD1)	Previously treated unresectable BTC	II	Recruiting	ORR, AEs, PFS, DOR, DCR, OS	Active
NCT04450732	GQ1001	Previously treated, HER2-positive advanced solid tumors	I	Recruiting	DLT, AEs, maximum tolerated dose (MTD)	Active
NCT02451553	Afatinib dimaleate Capecitabine	Previously treated, advanced refractory solid tumors, pancreatic cancer, or biliary cancer	I	Completed	DLT, AEs, MTD	Completed
NCT04660929	CT-0508 (CAR macrophages)	Previously treated, HER2-overexpressing solid tumors	I	Recruiting	AEs, ORR, PFS	Active
NCT04579380	Tucatinib Trastuzumab	Previously treated, locally advanced unresectable or metastatic solid tumors driven by HER2 alterations	II	Recruiting	cORR, DCR, PFS, OS, AEs	Active
NCT04278144	BDC-1001 (immune stimulating antibody conjugate, ISAC) Nivolumab	HER2-overexpressing advanced malignancies	I/II	Recruiting	AEs, DLT, ORR, DOR, DCR, PFS	Active
NCT04460456	SBT6050 (ISAC) Pembrolizumab Cemiplimab	HER2 expressing or amplified advanced malignancies	I	Active, not recruiting	DLT, AEs, ORR, DOR, DCR, PFS	Active, not recruiting
NCT05150691	DB-1303	HER2 expressing advanced solid tumors	I/II	Recruiting	DLT, ORR, AEs	Active
NCT00478140	Trastuzumab	HER2/neu-positive advanced gallbladder or biliary tract cancer	II	Terminated (Due to slow accrual)	ORR, DCR, OS	Terminated
NCT02999672	Trastuzumab emtansine	HER2 overexpressing solid tumors	II	Completed	OS, PFS, AEs	Completed
NCT04430738	Tucatinib Trastuzumab FOLFOX CAPOX Pembrolizumab	HER2+ gastrointestinal cancers	I/II	Recruiting	DLT, AEs, ORR, DOR, PFS, OS	Active
NCT03613168 (BILHER)	Trastuzumab GEMCIS	HER2+ biliary tract cancer	II	Completed	RR, PFS, OS	Completed
NCT00101036	Lapatinib ditosylate	Locally Advanced or Metastatic Biliary Tract or Liver Cancer That Cannot Be Removed By Surgery	II	Completed	ORR, OS, PFS	Completed

## Data Availability

Not applicable.

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
