# Peer review of "The Role of HER2 Status in the Biliary Tract Cancers"

_cancers, 2023, doi:10.3390/cancers15092628_

Round 1

Reviewer 1 Report

The authors reviewed and discussed the efficacy of HER-2 blocking antibodies or drug conjugates in HER-2 altered and/or amplified BTCs. I have read it with great interests. The manuscript was excellent. It was helpful for me and it may be very educational for other readers. I have no additional comments.

Author Response

We greatly appreciate your positive comments.

Reviewer 2 Report

This is a review paper regarding chemotherapy for BTCs including intrahepatic, extrahepatic, and gallbladder carcinomas, focusing mainly on HER2-targeted therapy. This paper comprehensively reviewed (1) the studies on the prognosis difference between patients with HER2 alterations and without, (2) the studies on the types and their frequencies of HER2 alterations according to the BTC sites, (3) the clinical trials and studies on HER2 targeted agents. 

 Although I have no specific requests to the authors, however, if possible, some more effort towards more conciseness is desired, e.g); making a short summary sentence of each study ahead of each introductory content.    

Author Response

Reviewer 2

This is a review paper regarding chemotherapy for BTCs including intrahepatic, extrahepatic, and gallbladder carcinomas, focusing mainly on HER2-targeted therapy. This paper comprehensively reviewed (1) the studies on the prognosis difference between patients with HER2 alterations and without, (2) the studies on the types and their frequencies of HER2 alterations according to the BTC sites, (3) the clinical trials and studies on HER2 targeted agents.

Although I have no specific requests to the authors, however, if possible, some more effort towards more conciseness is desired, e.g); making a short summary sentence of each study ahead of each introductory content.   

Author Response: Thank you for your feedback on our manuscript. We appreciate your suggestion regarding the need for more conciseness in the manuscript. We have taken your comment into consideration and made necessary revisions to improve the clarity of our introduction by adding a short summary sentence ahead of each study description.

We have included the following sentences to expand the introduction (Line 43-46):

“…In an era of immune-based therapies, the TOPAZ-1 trial evaluated the efficacy of durvalumab (anti-PDL1) plus GEMCIS in patients with BTC (NCT03875235). Addition of durvalumab to standard-of-care treatment significantly improved overall survival rate in patients with BTC.”

We have included the following sentences to expand the introduction (Line 77-80):

“…After the approval of a FGFR1-3 inhibitor by the FDA for the treatment of BTC, several other FGFR inhibitors were tested for their effectiveness. As a result, clinical trials were conducted to evaluate the efficacy of infigratinib and futibatinib.”

Reviewer 3 Report

Thank you for submitting your study entitled “The Role of HER2 Status in the Biliary Tract Cancers  in Cancers. This paper applied intermittent superior mesenteric artery occlusion (ISMAO) as a blood flow codiscuss the potential therapeutic utilization of HER2- 14 blocking antibodies and ongoing clinical trials assessing the efficacy of HER2-blocking antibodies 15 in HER2 altered and/or amplified biliary tract cancers. 

At this stage, it has some minor flaws or doubts that need to be addressed by the author.

1. Cholangiocarcinoma (CCA) constitutes a diverse group of malignancies emerging in the biliary tree. CCAs are divided into three subtypes depending on their anatom- ical site of origin: intrahepatic (iCCA), perihilar (pCCA) and distal (dCCA) CCAIs it appropriate for the author to include Ampulla of Vater into biliary tract tumors in Table 1 of the manuscript?

2. The role of human epidermal growth factor receptor 2 (HER2) in biliary tract tumors has been extensively discussed in the manuscript. Whether the author has considered the differences between the Eastern and Western races in the expression of HER2, I hope the author will have a proper discussion in this regard.

3. Clinically, biliary tumours have a poor prognosis, limited treatment options, varying effects of chemotherapy, and limited therapeutic targets. The use of HER2 as a therapeutic target and for biliary tumours has indeed been reported in the literature, but whether patients benefit from it is debatable. Therefore, treatment of biliary tumours should be combined with multi-target integrated therapy, where single drug use is difficult to break through in terms of efficacy, and the authors should discuss this aspect.

Minor comments 

In many places throughout the manuscript (too many to list), the language structure/grammar is in need of re-editing.

Author Response

Reviewer 3

Thank you for submitting your study entitled “The Role of HER2 Status in the Biliary Tract Cancers” in “Cancers”. This paper applied intermittent superior mesenteric artery occlusion (ISMAO) as a blood flow codiscuss the potential therapeutic utilization of HER2- 14 blocking antibodies and ongoing clinical trials assessing the efficacy of HER2-blocking antibodies 15 in HER2 altered and/or amplified biliary tract cancers.

At this stage, it has some minor flaws or doubts that need to be addressed by the author.

Author Response: We would like to express our gratitude for taking the time to review our article. Your comments and feedback are highly valuable to us, and we appreciate your effort in providing them. We have carefully considered your suggestions and have made necessary revisions to address them.

  1. Cholangiocarcinoma (CCA) constitutes a diverse group of malignancies emerging in the biliary tree. CCAs are divided into three subtypes depending on their anatomical site of origin: intrahepatic (iCCA), perihilar (pCCA) and distal (dCCA) CCA. Is it appropriate for the author to include Ampulla of Vater into biliary tract tumors in Table 1 of the manuscript?

Author Response: We concur that including the ampulla of water is crucial. As for the initial Table 1, we have already appended the Ampulla of Vater at the table's.

Primary tumor site

Frequency of HER2 alteration

Intrahepatic cholangiocarcinoma

5.8%

Extrahepatic cholangiocarcinoma

13.9%

Gallbladder cancer

36.4%

Ampulla of Vater

18.2%

Table 1. The frequency of HER2 aberrations in South Korean patients with BTC (adapted from [17]])

  1. The role of human epidermal growth factor receptor 2 (HER2) in biliary tract tumors has been extensively discussed in the manuscript. Whether the author has considered the differences between the Eastern and Western races in the expression of HER2, I hope the author will have a proper discussion in this regard.

Author Response: We appreciate the suggestion to consider the differences in HER2 expression between Eastern and Western races to include a proper discussion of this aspect in the manuscript. Thus, we propose to incorporate retrospective studies that compare HER2 expression levels across different racial backgrounds. While our primary aim was to provide a comprehensive review of clinical trials and the prognostic significance of HER2 expression, we agree that additional information will enhance the clarity of our manuscript. Nevertheless, the percent of HER2 overexpression by ethnicity vary between studies.

We have expanded the manuscript with following sentences (Line 153-162):

“…Moreover, there are retrospective studies investigated the differences in HER2 overexpression in Asian and Caucasian patients with biliary tract cancer. A meta-analysis revealed that HER2 expression was more prevalent in Asian patients (28.4%) than Western patients (19.7%) [18]. In a study from Japan, 454 cases of biliary tract cancer have been assessed for HER2 positivity. It has been demonstrated that HER2 positivity differed among different subtype of BTCs (3.7% in iCCA, 3% in perihilar eCCA, 18.5% in distal eCCA, 31.3% in GBC, and 16.4% in ampullary cancer) [19] The percentage of HER2 overexpression observed in different studies may differ because of varying factors such as the use of different cutoff values to determine overexpression, diverse detection methods employed, and the specific site of the primary tumor.”

  1. Clinically, biliary tumors have a poor prognosis, limited treatment options, varying effects of chemotherapy, and limited therapeutic targets. The use of HER2 as a therapeutic target and for biliary tumors has indeed been reported in the literature, but whether patients benefit from it is debatable. Therefore, treatment of biliary tumors should be combined with multi-target integrated therapy, where single drug use is difficult to break through in terms of efficacy, and the authors should discuss this aspect.​

Author Response: We agree that biliary tumors have a dismal prognosis, and limited treatment options with varying effects of chemotherapy. We have highlighted in our review the potential use of HER2 as a therapeutic target for biliary tumors, and its limited clinical benefit as a single agent.

Furthermore, we acknowledge that multi-target integrated therapy could be a promising approach to overcome the limitations of single-drug therapy. We have revised our manuscript to address this aspect and discuss the potential advantages of multi-target therapy in the treatment of biliary tumors.

We have expanded the manuscript with following sentences (Line 386-391):

“…Collateral and vertical oncogenic signaling pathways are the current hurdle in precision medicine. Even though results from clinical trials assessing HER2-targeting therapies in BTC are promising, the combinational treatment options might be imperative to achieve breakthrough responses. In this regard, combining immunotherapy or chemotherapy agents with anti-HER2 therapies seems to be feasible for greater and more durable responses [49].”

We have also stated that chemotherapy alone may not provide a significant survival advantage, and the addition of HER2-targeted therapies could be a potential alternative (Line 297-299).

“…Given the historical data demonstrating a 5% ORR with FOLFOX as second-line treatment for BTC [5], the promising results of this phase 2 study pave the way for future combination treatments utilizing HER2-targeted therapies.”

Minor comments

In many places throughout the manuscript (too many to list), the language structure/grammar is in need of re-editing.

Author Response: We appreciate your constructive feedback on the language structure and grammar of our manuscript. To improve the language and grammar of our manuscript, we have allocated additional time to carefully edit and proofread the manuscript before resubmission.

We appreciate you bringing this matter to our attention, and we are committed to ensuring that the updated version of our manuscript meets your expectations. In response, we have not only made several minor edits, but we have also revised certain sentences to provide further clarification. (Line 133-139):

“…The median overall survival (OS) in various types of biliary tract cancers (BTCs) differed significantly, as indicated by this study. In particular, ampullary tumors had not yet reached the median OS, intrahepatic tumors had a median OS of 55.3 months, extra-hepatic tumors had a median OS of 34.7 months, and gallbladder tumors had a median OS of 18.1 months. It is worth noting that these differences suggest that the primary site of BTC may be an important factor in determining patient prognosis.”

We have revised following sentences for further clarification (Line 180-184):

“…Due to the widespread acceptance of tumor agnostic approaches in precision medicine, there is currently significant interest in investigating whether anti-HER2 therapies can be applied to other types of cancer, including biliary tract cancer (BTC). This may have important implications for the development of targeted therapies for BTC and other cancers with HER2 aberrations.”